# A Water-Dispersible Carboxylated Carbon Nitride Nanoparticles-Based Electrochemical Platform for Direct Reporting of Hydroxyl Radical in Meat

**DOI:** 10.3390/foods11010040

**Published:** 2021-12-24

**Authors:** Tingting Han, Yang Huang, Chong Sun, Daoying Wang, Weimin Xu

**Affiliations:** 1Institute of Agricultural Products Processing, Jiangsu Academy of Agricultural Sciences, Nanjing 210014, China; 2020015041@qymail.bhu.edu.cn (T.H.); 192702014@njnu.edu.cn (Y.H.); xuweimin@jaas.ac.cn (W.X.); 2Jiangsu Key Laboratory for Food Quality and Safety-State Key Laboratory Cultivation Base, Ministry of Science and Technology, Nanjing 210014, China

**Keywords:** carboxylated-*g*-C_3_N_4_ NPs, hydroxyl radical, electrochemical sensor, food safety

## Abstract

In this paper, carboxylated carbon nitride nanoparticles (carboxylated-*g*-C_3_N_4_ NPs) were prepared through a one-step molten salts method. The synthesized material was characterized by transmission electron microscope (TEM), Fourier transform-infrared spectra (FTIR), and X-ray photoelectron spectroscopy (XPS), etc. An electrochemical sensor based on single-stranded oligonucleotide/carboxylated-*g*-C_3_N_4_/chitosan/glassy carbon electrode (ssDNA/carboxylated-*g*-C_3_N_4_/chitosan/GCE) was constructed for determination of the hydroxyl radical (^•^OH), and methylene blue (MB) was used as a signal molecule. The sensor showed a suitable electrochemical response toward ^•^OH from 4.06 to 122.79 fM with a detection limit of 1.35 fM. The selectivity, reproducibility, and stability were also presented. Application of the sensor to real meat samples (i.e., pork, chicken, shrimp, and sausage) was performed, and the results indicated the proposed method could be used to detect ^•^OH in practical samples. The proposed sensor holds a great promise to be applied in the fields of food safety.

## 1. Introduction

Numerous studies demonstrate that color, odor, flavor, and nutritional ingredients are the key indicators in assessing the quality of meat [1]. These indicators are closely associated with the process of oxidation, which acts as one of the main causes of spoiled meat that generally occurs in lipid and protein [2]. During the metabolic oxidation process, the molecule oxygen participates in a series of metabolic reactions. Approximately 98% of oxygen is combined with glucose and fat in the organelle, thus is converted into energy to meet the needs of cellular activity. Another 2% of oxygen leads to the generation of free radicals in the form of reactive oxygen species (ROS) [3]. Moreover, ROS can be released by a series of chemical and enzymatic processes during slaughtering, cutting, processing, packaging, and cooking [4]. ROS, including superoxide anion (O_2_^•−^), singlet oxygen (^1^O_2_), 1,1-diphenyl-2-trinitrophenylhydrazine radical (DPPH^•^), hydroxyl radical (^•^OH) and so on, can interact with a variety of species due to their high activity, which causes a series of destructive chain reactions to cells [5]. Among ROS, ^•^OH is considered as the most active and harmful free radical, which reacts with many biological species such as sugars, amino acids, phospholipids, nucleic acids, and organic acids. These reactions cause lipid and protein peroxidation [6], leading to the loss of some essential amino acids and the generation of some toxic substances in meat products [7]. In addition, the oxidation stress induced by high levels of ^•^OH would cause apoptosis, mutation, and many diseases such as hypertension, arteriosclerosis, diabetes mellitus, cataract, arthritis, cancer, etc. [8]. Consequently, there is a great demand to find an appropriate approach for rapid and sensitive determination of ^•^OH.

In the past few decades, a number of methods have been developed and applied for detection of ^•^OH, including electron spin resonance (ESR) [9], chemiluminescence [10], gas chromatography/mass spectrometry (GC-MS) [11], capillary electrophoresis (CE) [12] and fluorescence [13]. Compared with these methods, electrochemical sensors have presented many advantages such as low detection limit, simplicity of preparation, and rapid response time [14]. Xu et al. developed a novel electrochemical sensor based on 6-(Ferrocenyl) hexanethiol (6-FcHT) self-assembled nanoporous gold layer modified gold electrode (6-FcHT/NPGL/GE) for detection of the release of ^•^OH from living cells [15]. Duanghathaipornsuk et al. constructed an efficient sensing device based on cerium oxide nanoparticles and graphene oxide (CeNP/GO) composite for the detection of ^•^OH [16]. However, developing a highly selective and sensitive electrochemical sensing platform to detect free radicals remains a challenging work due to uncontrollable reactivity, extreme short lifetime, and high reaction rate constant [17].

Graphitic carbon nitride (*g*-C_3_N_4_), composed of repeating tri-s-triazine units, is the most stable allotrope among diverse carbon nitride materials under air conditions [18]. Much attention has been paid to *g*-C_3_N_4_ due to the unique structure and characteristics, including high thermal and chemical stability, nontoxicity, suitable biocompatibility, special electronic and optical properties, which has been extensively applied in many areas such as bioimaging [19], surface-enhanced Raman scattering sensing [20], electrochemical sensor [21], photocatalysis [22] and solar-driven water-splitting [23]. Despite prominent features of *g*-C_3_N_4_, the application of bulk polymeric carbon nitride (bulk-*g*-C_3_N_4_) has been confined to a certain extent for low specific surface area, poor water dispersion, and weak electrical conductivity [24]. Compared with bulk-*g*-C_3_N_4_, carboxylated-*g*-C_3_N_4_ nanoparticles (NPs) prepared have a larger surface area, more active sites, better electron transfer capacity, and better water dispersion due to the presence of C=O and OH groups.

In this paper, carboxylated-*g*-C_3_N_4_ NPs were successfully prepared by a molten salt method, using melamine as the precursor and NaCl/KCl as eutectic salts. Then, a single-stranded oligonucleotide/carboxylated-*g*-C_3_N_4_/chitosan/glassy carbon electrode (ssDNA/carboxylated-*g*-C_3_N_4_/chitosan/GCE) based on Fenton-mediated ssDNA damage mechanisms for sensitive determination of ^•^OH was constructed, and methylene blue (MB) was used as a signal probe. It was confirmed that ^•^OH could attack ssDNA to cause several ways of damage, including strand breakage and base release [25]. The prepared sensor presented a wide linear range and suitable selectivity toward the detection of ^•^OH. The application of the sensor to the determination of ^•^OH in real meat samples was tested and compared with that of ESR. More details were discussed in the following sections.

## 2. Materials and Methods

### 2.1. Materials and Reagents

ssDNA was purchased from Sangon Biotech Co., Ltd. (Shanghai, China), and the sequence is as follows: 5′-NH_2_-(CH)_6_-GGT CCG CTT GCT CTC GC-3′. Melamine (≥99%), potassium chloride (KCl, 99.5%), sodium chloride (NaCl, 99.9%), hydrochloric acid (HCl, 36%~38%), N-hydroxysuccinimide (NHS), 1-ethyl-3-(3-dimethylaminopropyl) carbodiimide hydrochloride (EDC), FeSO_4_·7H_2_O (99%~101%), H_2_O_2_ (30%), MB, chitosan, acetic acid, and xanthine were all obtained from Aladdin (Shanghai, China). Xanthine oxidase was purchased from Shanghai Yuanye Bio-Technology Co., Ltd. 5-*tert*-Butoxycarbonyl-5-methyl-1-pyrroline-N-oxide (BMPO) was purchased from Dojindo Laboratories (Japan). BMPO was dissolved in ultra-pure water and was used as the trapping agent to radicals. The final concentration of BMPO in samples was 10 mM. Phosphate buffer solution (PBS) was prepared by mixing Na_2_HPO_4_ and KH_2_PO_4_. The mixture of a series concentration of Fe^2+^ and H_2_O_2_ (the concentration ratio was 1:6) promoted the formation of ^•^OH. ^1^O_2_ was obtained by the reaction of NaClO and H_2_O_2_ (1 mM NaClO and 1 mM H_2_O_2_). O_2_^•−^ was generated by a mixture of 100 µM xanthine and 22 mU xanthine oxidase. All chemicals were analytical reagents, and all solutions were prepared with ultra-pure water.

### 2.2. Synthesis of Carboxylated-g-C_3_N_4_ NPs

The carboxylated-*g*-C_3_N_4_ NP_S_ were synthesized according to the molten method with some modifications [26]. A total of 8.00 g melamine was mixed with 2.50 g NaCl and 3.12 g KCl. Then, the mixture was well ground in an agate mortar for 1 h until it turned into a uniform powder. The powder was placed into a muffle furnace and retained at 670 °C for 4 h with a heating rate of 10 °C·min^−1^. A bright yellow melting block was obtained after the products naturally cooled down to room temperature. The separation process of the products was as follows: Firstly, the melting block was ground into a uniform powder. Then, the salts in the products were removed by the diluted hydrochloric acid, which contributed to the aggregation of water-dispersible nano-sized carbon nitride at the same time. After centrifugation, the products were dispersed in ultra-pure water and centrifugated once more to remove the large carbon nitride particles. The supernatant was dialyzed through a dialysis bag (MWCO = 500) in ultra-pure water until the pH was neutral and the remnant Na^+^, K^+^, Cl^−^ was virtually removed. Finally, the obtained solution was lyophilized, and the carboxylated-*g*-C_3_N_4_ NP_S_ were obtained. The as-synthesized carboxylated-*g*-C_3_N_4_ NP_S_ could be re-dispersed in ultra-pure water for the following experiments. Meanwhile, the preparation of bulk-*g*-C_3_N_4_ was carried out. Melamine was put into a muffle furnace and was heated to 550 °C with a rate of 10 °C·min^−1^, and remained for 4 h. The bulk-*g*-C_3_N_4_ was obtained after it cooled to ambient temperature.

### 2.3. Apparatus and Instruments

The morphology of bulk-*g*-C_3_N_4_ and carboxylated-*g*-C_3_N_4_ NP_S_ was observed with an H-7650 transmission electron microscope (TEM, HITACHI, Tokyo, Japan) performed at an accelerating voltage of 80 kV. Fourier transform-infrared spectroscopy (FTIR, VARIAN Cary 5000, Santa Clara, CA, USA), zeta potential (ZP, Nano ZS90, Malvern Instruments, Worcestershire, UK), and X-ray photoelectron spectroscopy (XPS, Thermo ESCALAB250Xi, Waltham, MA, USA) were used to characterize the successful synthesis of the carboxylated-*g*-C_3_N_4_ NPs. The ESR measurements were made at room temperature with a Bruker EMX 10/12 X-band ESR spectrophotometer (Bruker, Rheinstetten, Germany). Microwave power 20 mW, sweep width 200 Gauss, modulation frequency 100 kHz, modulation amplitude 1.0 Gauss, sweep time 83.89 s, conversion time 81.92 ms, time constant 40.96 ms, and 3 scans per sample were accumulated. The software WINEPR version 4.40.11.65 was used to analyze the spectra. All electrochemical measurements were carried out at a CHI760E electrochemical workstation (Shanghai Chenhua Instruments Co., Ltd. Shanghai, China) with a conventional three-electrode system. The GCE or modified electrode was used as a working electrode, a saturated calomel electrode (SCE) was used as the reference electrode, and a platinum wire electrode was employed as the counter electrode. Meat samples were crushed and homogenized with an Ultra Turrax (T25, IKA, Staufen, Germany). A centrifuge (Allegra 64R, Beckman, Brea, CA, USA) was used to collect supernatant.

### 2.4. Fabrication of MB/ssDNA/Carboxylated-g-C_3_N_4_/Chitosan/GCE

GCEs (the apparent surface of the electrode is estimated to be 0.078 cm^2^) were polished with 0.3 and 0.05 µm aluminum oxide powder and then sonicated in ethanol and ultra-pure water, respectively, several times. After that, GCEs were dried under nitrogen gas. The fabrication process of the sensor is displayed in Figure 1. Firstly, the carboxylated-*g*-C_3_N_4_ NPs (5 µL, 3 mg·mL^−1^) and chitosan (5 µL, 1% dissolved in acetic acid) were mixed and dropped onto the surface of GCE. Chitosan was chosen as an adhesive molecule to immobilize the carboxylated-*g*-C_3_N_4_ NPs on the clear surface of GCE. Then, the modified GCE was immersed in 40 µL EDC/NHS (1:1) solution to activate the carboxyl group of the carboxylated-*g*-C_3_N_4_ NP_S_. The carboxylated-*g*-C_3_N_4_/chitosan/GCE electrode was rinsed with ultra-pure water in order to remove extra EDC/NHS solution. After that, 5.0 µL amino-terminated ssDNA solution was dropped on the surface of carboxylated-*g*-C_3_N_4_/chitosan/GCE and incubated for 1 h to combine carboxyl and amino completely. Then, the ssDNA/carboxylated-*g*-C_3_N_4_/chitosan/GCE were rinsed with ultra-pure water to remove uncombined ssDNA. For determination of ^•^OH, the modified electrode was immersed in Fenton solution (c(Fe^2+^): c(H_2_O_2_) = 1:6, pH = 3.5) for 30 min. The ssDNA chain could be broken by ^•^OH, which was produced by Fenton reaction. After that, the ssDNA/carboxylated-*g*-C_3_N_4_/chitosan/GCE was washed with ultra-pure water to remove the broken ssDNA strand and dried at room temperature. Finally, the ssDNA/carboxylated-*g*-C_3_N_4_/chitosan/GCE was immersed in PBS containing 40 μM MB. After combination with MB, the MB/ssDNA/carboxylated-*g*-C_3_N_4_/chitosan/GCE was washed with ultra-pure water several times to remove uncombined MB.

### 2.5. Electrochemical Measurements

Cyclic voltammetry (CV) experiments were performed at 1 mM [Fe(CN)_6_]^3−/4−^ (1:1) solution containing 0.1 M KCl with a scan rate of 100 mV·s^−1^, and the potential range was from −0.4 to 0.7 V. Electrochemical impedance spectroscopy (EIS) assessments were made in 5 mM [Fe(CN)_6_]^3−/4−^ solution containing 0.1 M KCl with an AC voltage of 5 mV amplitude and the frequency range was range from 1 to 10^5^ Hz. Square wave voltammetry (SWV) was carried out over the potential range from −0.55 to −0.05 V. Each test was repeated three times, and all electrochemical measurements were carried out at 25 °C in a digital biochemical incubator.

### 2.6. Detection of Real Samples

Pork, chicken, shrimp, and sausage were purchased from a local supermarket. ^•^OH was extracted according to the method described as follows [27]: 2 g of each type of meat were crushed and homogenized in 10 mL 0.1 M PBS (pH = 7.4) at 6000 r·min^−1^ for 60 s. The homogenate was adjusted to pH = 3.5 with 1 M H_2_SO_4_, and 20 µL of 100 mM BMPO was added into the homogenate. Then, different concentrations of Fe^2+^ and H_2_O_2_ were added into the homogenate and put into an ice-water bath for 10 min. After that, the homogenate was centrifuged at 12,000× *g* for 15 min at 10~15 °C. Then, the supernatant was collected and used for the biochemical assay. The contrast experiment used to detect ^•^OH was conducted by ESR spectroscopy. A linear relationship between ESR peak area and the concentration of ^•^OH was obtained. Then, the ^•^OH levels were quantified by the peak areas.

## 3. Results and Discussion

### 3.1. Characterization of the Carboxylated-g-C_3_N_4_ NPs

The morphology and size of bulk-*g*-C_3_N_4_ and carboxylated-*g*-C_3_N_4_ NPs were characterized by TEM images (Figure 1). It could be observed that the bulk-*g*-C_3_N_4_ (Figure 1A) was flake-like and exhibited a two-dimensional plane structure with the size of a few microns [28]. In contrast to bulk-*g*-C_3_N_4,_ the carboxylated-*g*-C_3_N_4_ NPs were nanoscale and dispersed uniformly, and the particle size was approximately 20 nm (Figure 1B). The result could be ascribed to the highly reactive atmosphere of molten salt in which salty ions migrated quickly and inserted into the structure of polymeric carbon nitride, causing the breakage of bulk-*g*-C_3_N_4_ [29].

FTIR spectroscopy and ZP distribution, as representative tools, were used to verify whether the synthesis process of carboxylated-*g*-C_3_N_4_ NPs was successful. As demonstrated in Figure 1C, the peak at around 810 cm^−1^ originated from out-plane flexural vibration, which was the characteristic infrared absorption peak of the triazine unit [30]. The absorption bands at 1480 and 1634 cm^−1^ were caused by the stretching vibration of C–N and C=N in heptazine. The peak at 1706 cm^−1^ corresponded to the bending mode of C=O, and the absorption peaks at about 1393 and 1580 cm^−1^ were the characteristic bands of -COO, demonstrating carbon nitride is functionalized with the carboxyl group successfully [31]. The band at around 2214 cm^−1^ arose from the stretching vibration of C≡N, and the appearance of cyano groups might be ascribed to the cleavage of bulk carbon nitride in the molten salt environment [32]. The broadband between 3000 and 3500 cm^−1^ indicated the existence of –OH and –NH groups, –OH assigned to the carboxyl group, and –NH was related to the primary and secondary amines (and their intermolecular hydrogen bonding) [33]. As displayed in Figure 1D, the ZP of bulk-*g*-C_3_N_4_ was −0.056 mV, which was almost zero (top). However, the ZP of carboxylated-*g*-C_3_N_4_ NPs was −24.0 mV (bottom), indicating carboxylated-*g*-C_3_N_4_ NPs were negatively charged [34]. These results demonstrated that the carboxylated-*g*-C_3_N_4_ NPs were synthesized successfully.

The element chemical states and binding structure of the carboxylated-*g*-C_3_N_4_ NPs were studied by XPS. The results indicated that the carboxylated-*g*-C_3_N_4_ NPs mainly contained C, N, and O elements (Figure 2A). There were three peaks in the high-resolution C 1 s spectrum (Figure 2B). The peak at 288.8 eV originated from sp^2^ carbon atoms in the form of C−N=C and C=O [29]. The peaks located at 286.7 and 284.7 eV were relevant to the carbon atom that existed in C≡N [35] and the standard reference carbon [36]. The N 1s spectra presented three peaks with the energy of 398.9, 399.9, and 400.8 eV (Figure 2C). The peaks at 398.9 and 399.9 eV were attributed to the sp^2^ N atom involved in C–N=C (triazine ring) and C≡N functional group, respectively. The peak situated at 400.8 eV was ascribed to the N–(C)_3_ group in which a nitrogen atom was combined with three carbon atoms, indicating that the carboxylated-*g*-C_3_N_4_ NPs still retained a tri-s-triazine structure but not repeated triazine units [26]. The high-resolution O 1 s spectrum (Figure 2D) was divided into two peaks, located at 531.9 and 533.4 eV, corresponding to the groups of C=O and C–OH [29,37]. These results obtained by XPS were in step with that of the FTIR spectrum.

### 3.2. The Mechanism of Detection ^•^OH

The detection of ^•^OH was performed at ssDNA/carboxylated-*g*-C_3_N_4_/chitosan/GCE using MB as a signal probe. The Fenton reaction was used as an important pathway to produce ^•^OH. Different concentrations of H_2_O_2_ were added into ferrous solution to generate ^•^OH. The reaction principle was as follow: Fe^2+^ + H_2_O_2_ ⟶ Fe^3+^ + (OH)^−^ + ^•^OH. In addition, ^•^OH could cause ssDNA strand breakage, and more ^•^OH would cause more ssDNA strand breakage. The interaction between ssDNA and MB was attributed to electrostatic binding, and the SWV response signal became small with MB decreasing. From Appendix A, a reduction peak of MB was obtained from the ssDNA/carboxylated-*g*-C_3_N_4_/chitosan/GCE (curve a) without any treatment. When the electrode was incubated with 1 mM Fe^2+^ (curve b) or 6 mM H_2_O_2_ (curve c), there was no obvious change in peak current. The result indicated that Fe^2+^ or H_2_O_2_ individually could not cause ssDNA breakage. However, the reduction peak current decreased dramatically when the electrode was immersed in 1 mM Fe^2+^ and 6 mM H_2_O_2_ (curve d), owing to the fact that the generation of ^•^OH would induce ssDNA damage. These results demonstrated that the fabricated sensor could be used to detect ^•^OH successfully.

### 3.3. Optimization of Experimental Parameters

In order to possess suitable electrochemical performance, the concentration of the carboxylated-*g*-C_3_N_4_ NPs modified on the GCE was optimized, and the results obtained by the linear scan voltammetry (LSV) method were represented in Appendix A. It could be seen that the current signals increased with the concentrations of carboxylated-*g*-C_3_N_4_ NPs ranging from 1 to 3 mg·mL^−1^ and decreased when the concentration was over 3 mg·mL^−1^. This was because higher concentrations of carboxylated-*g*-C_3_N_4_ NPs were apt to aggregate on the surface of the electrode, leading to poor conductivity. Thus, 3 mg·mL^−1^ was chosen for the following experiments.

The parameters, including the concentrations of ssDNA and MB, the incubation time of carboxylated-*g*-C_3_N_4_ NPs and amino-terminated ssDNA, the combining time of ssDNA and MB, were studied by the SWV method. The current signals corresponding to different concentrations of ssDNA are presented in Figure 3. It could be observed from Figure 3A that the peak current increased with the concentration of ssDNA ranging from 1 to 4 μM and remained stable when the concentration was over 4 μM, indicating that the carboxyl group of the carboxylated-*g*-C_3_N_4_ NPs was combined completely with amino-terminated ssDNA. Therefore, 4 μM was selected as the best concentration of ssDNA. Moreover, the concentration of MB was investigated. As shown in Figure 3B, the current went up with the increasing concentration of MB, achieved maximum at the concentration value of 40 μM, and then kept stable. Hence, 40 μM was considered as the optimal concentration of MB. Figure 3C represented the relationship between the current signal and incubation time of carboxylated-*g*-C_3_N_4_ NPs and amino-terminated ssDNA. It was found that the longer the incubation time, the larger the current signal value was obtained, and the current held steady after 40 min. As a result, 45 min was identified as the optimal incubation time. With regard to the binding time of ssDNA and MB, the peak current increased and remained stable after 20 min, suggesting that ssDNA was bound with MB entirely (Figure 3D). Thus, 20 min was chosen. To sum up, the above optimization conditions were employed in the following experiments.

Correlation is a statistical technique used to calculate Pearson’s correlation coefficient (r) between independent and dependent variables [38]. The correlation of the above parameters was evaluated by using IBM SPSS Statistics 25. As shown in Appendix A, all the parameters investigated had a positive correlation with the electrochemical signal. In addition, there was a higher relationship of 0.935 (*p* ≤ 0.01) between the binding time of ssDNA and MB with the electrochemical signal, which indicated that the binding time of ssDNA and MB was a major factor affecting the electrochemical signal in the construction process of sensor. Furthermore, there were several strong correlations between parameters, such as the concentrations of ssDNA and the binding time of ssDNA and MB (r = 0.998, *p* ≤ 0.01). These are attributed to the fact that a higher concentration of ssDNA on the electrode surface could be more accessible in binding with MB. The correlation analysis verified that these parameters chosen most strongly affected the electrochemical performance of the sensor, and the optimization of experimental parameters contributed to enhancing sensitivity.

### 3.4. Electrochemical Characteristics of the Fabricated Electrode

The electrochemical behavior of the chitosan/GCE (curve a, Figure 4A), bulk-*g*-C_3_N_4_/chitosan/GCE (curve b, Figure 4A), carboxylated-*g*-C_3_N_4_/chitosan/GCE (curve c, Figure 4A), and ssDNA/carboxylated-*g*-C_3_N_4_/chitosan/GCE (curve d, Figure 4A) were investigated by CV technique. Compared with chitosan/GCE, the peak current increased slightly after bulk-*g*-C_3_N_4_ modified GCE. This could be ascribed to the weak conductivity of the bulk-*g*-C_3_N_4_ [39]. After the carboxylated-*g*-C_3_N_4_ NPs was coated on the chitosan/GCE, the peak current increased significantly, owing to the fact that the carboxylated-*g*-C_3_N_4_ NPs possessed many edge defects, which could accelerate electron transfer [40]. After incubation with ssDNA, the peak current of ssDNA/carboxylated-*g*-C_3_N_4_/chitosan/GCE decreased remarkably, which could be attributed to the introduction of non-conducting ssDNA, which caused steric hindrance on the GCE surface. The results indicated that ssDNA was successfully combined with the carboxylated-*g*-C_3_N_4_ NPs.

In order to further investigate the surface properties of chitosan/GCE, bulk-g-C_3_N_4_/chitosan/GCE, carboxylated-*g*-C_3_N_4_/chitosan/GCE, and ssDNA/carboxylated-*g*-C_3_N_4_/chitosan/GCE, EIS was conducted (Figure 4B). The semi-circle diameters of different modified electrodes in EIS are closely relevant to the efficiency of charge transfer [41]. The bulk-*g*-C_3_N_4_/chitosan/GCE showed a smaller diameter than chitosan/GCE because of its slightly weak electrical conductivity. However, the semi-circle diameter of the carboxylated-*g*-C_3_N_4_/chitosan/GCE reduced more remarkably than that of bulk-*g*-C_3_N_4_/chitosan/GCE because the surface of carboxylated-*g*-C_3_N_4_ NPs had many defects after calcination, leading to improvement of electron transfer efficiency. After incubation with ssDNA, the diameter of ssDNA/carboxylated-*g*-C_3_N_4_/chitosan/GCE became larger than carboxylated-*g*-C_3_N_4_/chitosan/GCE because ssDNA was non-conducting. These results were in agreement with data obtained from CV measurements.

### 3.5. Electrochemical Detection of ^•^OH

Different concentrations of ^•^OH were obtained by hydrolyzing the corresponding amount of Fe^2+^ and H_2_O_2_ for a constant time. The oxidation of sodium benzoate by ^•^OH would be from several oxidation products such as sodium salicylate, monohydroxybenzoates, dihydroxybenzoates, and phenol. Among them, sodium salicylate shows the highest fluorescence intensity, which is used to calculate the concentration of ^•^OH [6]. The steady-state concentration of ^•^OH is calculated according to the oxidation kinetics of sodium benzoate. A mixture solution containing 0.1 mM of sodium benzoate and a series concentration of Fe^2+^ (0.002–10 mM) and H_2_O_2_ (0.012–60 mM) was incubated for some time. After that, the fluorescence of the mixture solution was detected at 400 nm by being excited at 330 nm. The concentration of sodium salicylate in the mixture solution was calculated by the fitted linear equation y = 6.3923x − 3.854, R^2^ = 0.9960 (Appendix A), and then plotted versus the reaction time (Appendix A). The kinetics of benzoate oxidation followed the rate equation r = k_2_[^•^OH][benzoate], where variables r and k_2_ represent the oxidation rate and rate constant, respectively. The concentrations of ^•^OH were calculated by the initial instantaneous reaction rate and the rate constant k_2_ (5.9 × 10^9^ M^−1^·s^−1^) [42].

Under the optimized conditions, the reduction peak of MB was recorded by the SWV method in 0.1 M PBS (pH = 7.4). The relationship between peak currents and the concentration of ^•^OH was represented in Figure 4C. It showed that the peak current decreased gradually with the increasing concentration of ^•^OH. The linear regression equation between current difference (ΔI_p_) and the concentration of ^•^OH was ΔI_p_ (μA) = 0.300[^•^OH] − 0.0429 (R^2^ = 0.9933), ΔI_p_ = I_0_ − I_p_, I_0_ and I_p_ were the peak currents of MB obtained from electrodes before and after incubated with various concentration of ^•^OH (Figure 4D). The limit of detection was estimated from the equation: LOD = 3σ/K, σ was the standard deviation of the blank solution (i.e., without ^•^OH), and K was the slope of the calibration graph [43]. Thus, ΔI_p_ became larger for decreasing MB values. The developed method showed a linear range from 4.06 ×10^−15^ to 1.23 ×10^−13^ M, and the detection limit is 1.35 × 10^−15^ M (S/N = 3). As shown in Table 1, compared with other methods, the prepared sensor in this paper showed a lower linear range and detection limit [42,44,45,46,47]. The results proved that the carboxylated-*g*-C_3_N_4_ NP_S_ had a great effect on the electrochemical response of the electrode reaction for ^•^OH and provided a suitable microenvironment for ssDNA and MB to transfer electrons with underlying GCE.

### 3.6. Specificity, Stability, and Reproducibility of the Fabricated Sensor

Due to the presence of other ROS such as H_2_O_2_, O_2_^•**−**^ and ^1^O_2_, the specificity of the fabricated sensor was explored by the SWV method. The relative response in Figure 5 was obtained by signals of O_2_^•^^−^ (100 µM xanthine + 22 mU xanthine oxidase), ^1^O_2_ (1 mM NaClO and 1 mM H_2_O_2_) and H_2_O_2_ (1 mM) dividing the signal of ^•^OH (1 mM Fe^2+^ and 6 mM H_2_O_2_) then multiplying 100%. The results indicated that the sensor had a suitable selectivity for the determination of ^•^OH. The better selectivity of this sensor could be ascribed to the high-specificity of ^•^OH for ssDNA cleavage [48]. The stability of the electrode was investigated by analyzing the SWV signal. The current value maintained 92.05% and 85.56% after 3 and 7 days storage in a refrigerator at 4 °C, proving that the sensor possessed relatively suitable stability. The reproducibility of the sensor was estimated by using five electrodes to detect ^•^OH, and the results showed excellent reproducibility because the relative standard deviation was only 5.43%.

### 3.7. Real Sample Analysis

In order to verify the practicability of the sensor, different types of meat samples, including pork, chicken, shrimp, and sausage, were determined. The results of the determination are illustrated in Table 2. It was shown that the initial concentration of ^•^OH in meat could not be detected directly by the ESR method. This was ascribed to the fact that ^•^OH was in a rather low physiological concentration in fresh meats [49]. For the purpose of application of the sensor, ^•^OH was detected by the standard addition method. The concentrations of ^•^OH were calculated by the calibration curve in Figure 4D. Every experiment was repeated three times in parallel, and ESR was used as a reference method. As shown in Table 2, the recovery of the sensor ranged from 97.84% to 99.34%, with RSDs less than 6%. Meanwhile, there was no significant difference between the two methods, indicating that the proposed method can be used to detect ^•^OH in practical samples.

## 4. Conclusions

In summary, carboxylated-*g*-C_3_N_4_ NPs were prepared by the molten salt method. The carboxylated-*g*-C_3_N_4_ NPs had a larger surface area, more satisfactory electro-conductivity, and better dispersity compared with bulk-*g*-C_3_N_4_. Based on these properties, a novel and convenient sensor based on ssDNA/carboxylated-*g*-C_3_N_4_/chitosan/GCE for sensitive detection of ^•^OH was developed. Under the optimal conditions, there was a suitable linear relationship between peak current and the concentration of ^•^OH in the range from 4.06 to 122.79 fM with the detection limit of 1.35 fM. In fact, the developed sensor showed higher specificity, reproducibility, stability, and anti-interference ability toward detection of ^•^OH. Furthermore, the sensor could be applied in real meat samples, and the results were consistent with that of the ESR method. Hence, this work provided a promising sensor for sensitive detection of ^•^OH.

## Data Availability

The data presented in this study are available upon request from the corresponding author.

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
