# Peer review of "A Water-Dispersible Carboxylated Carbon Nitride Nanoparticles-Based Electrochemical Platform for Direct Reporting of Hydroxyl Radical in Meat"

_foods, 2021, doi:10.3390/foods11010040_

Round 1

Reviewer 1 Report

Minor remarks

  • All minor remarks are highlighted in the manuscript.

Major remarks

  • Each reference is desirable to be discussed separately. Avoid lumping the references in the Introduction.
  • Have in mind that the process optimization can be performed using the experimental design and similar mathematical tools. Using that approach, it is possible to analyze the simultaneous impacts of different parameters on the defined responses of the system, as well as the analysis of the interactions between them.

Author Response

Dear Editor,

Thank you very much for your letter and the referees’ reports. According to your comments and requirements, we have made modification on the original manuscript. Here, we have submitted the revised manuscript online for your approval. A revised manuscript with the correction sections colour marked was attached as the supplemental material and for easy check/editing purpose.

A document of Detailed Responses to Reviewers answering every question from the referees was also summarized and enclosed.

Thank you indeed for your time and efforts.

I’ m looking forward to some good news from you soon.

Best wishes,

Chong Sun

Institute of Agricultural Products Processing

Jiangsu Academy of Agricultural Sciences

Reviewer #1:

Each reference is desirable to be discussed separately. Avoid lumping the references in the Introduction.

Reply: Thanks for your comments. We have made appropriate modifications according to your suggestions. Please see the section of 1. Introduction in the revised manuscript.

Have in mind that the process optimization can be performed using the experimental design and similar mathematical tools. Using that approach, it is possible to analyze the simultaneous impacts of different parameters on the defined responses of the system, as well as the analysis of the interactions between them.

Reply: Thanks for your suggestions. New related discussion were added in the section of 3.3. Optimization of experimental parameters according to your suggstion. Please see the revised manuscript.

“Correlation is a statistical technique used to calculate Pearson’s correlation coefficient (r) between independent and dependent variables [38]. The correlation of the above parameters was evaluated by using IBM SPSS Statistics 25. As shown in Table S1, all the parameters investigated had a positive correlation with the electrochemical signal. In addition, there was a higher relationship of 0.935 (p≤0.01) between the binding time of ssDNA and MB with the electrochemical signal, which indicated that the binding time of ssDNA and MB was a major factor affecting the electrochemical signal in the construction process of sensor. Furthermore, there were several strong correlations between parameters such as the concentrations of ssDNA and the binding time of ssDNA and MB (r=0.998, p≤0.01). These attributed to the fact that a higher concentration of ssDNA on the electrode surface could be more accessible in binding with MB. The correlation analysis verified these parameters chosen most strongly affected the electrochemical performance of the sensor, and the optimization of experimental parameters was contribute to enhancing sensitivity.”

Table S1. Correlation coefficient matrix of sensor evaluation and experimental parameters. Positive coefficients indicated a direct relationship between variables in the matrix. *p≤0.05, **p≤0.01.

electrochemical signal

The concentrations of ssDNA

The concentrations of MB

The incubation time of carboxylated-g-C3N4 NPs and amino-terminated ssDNA

The binding time of ssDNA and MB

Electrochemical signal

1

The concentrations of ssDNA

0.890*

1

The concentrations of MB

0.864*

0.889*

1

The incubation time of carboxylated-g-C3N4 NPs and amino-terminated ssDNA

0.829*

0.919*

0.986**

1

The binding time of ssDNA and MB

0.935*

0.998**

0.907*

0.927*

1

Reviewer 2 Report

The  subject  presented in the manuscript “A Water-Dispersible Carboxylated-g-C3N4 Nanoparticles-Based Electrochemical Platform for Direct Reporting of Hydroxyl Radical in Meat” is interesting. Therefore, I recommend this manuscript should make minor revision, and some general comments/suggestions/questions are in the following: 
1) I recommend not using “g-C3N4” in the title, as it is difficult to read the term out loud. Maybe using “carboxylated carbon nitride nanoparticles” as defined in the Abstract would be a good choice, or something else that authors consider to be a better suit. I have used chemical formulas on my publications before, and I realized later that using a more general term would be easier to promote the paper and therefore increase the citations. Of course, it is just a suggestion based on my previous experience and may not be the rule. 
2) Line 44: Is there a typo on “uarthritis”? 
3) I suggest adding examples of other devices based on electrochemical measurements applied to the detection of the hydroxyl radical in the Introduction section, to complete the state-of-the-art. You may use this opportunity to emphasize what are the advantages of your platform compared to the others in the literature. 
4) Section 2.5: Is there a reason for conducting CV and EIS in different concentrations (1 mM and 5 mM, respectively)? 
5) Line 163: As your paper is intended to be accessed by a global audience, please specify which is the “ambient temperature”. 
6) The figures and table presented in the Supplementary Materials (or Supporting Information, as you named in the word file)  would  be  perfectly  fine in the main manuscript. Is there a limited number of figures in the Foods journal? Otherwise, I don’t see a reason why not add them in the 
main manuscript. The data is not repetitive information and the visual outcome of Figure S5, for instance, would highlight the selective detection of your device towards hydroxyl radical compared to others.  
7) Line 243: Please define LSV. 
8) Figure 3B: On the x-axis, I believe there is no minus sign, it should be positive Z’. Usually, the negative is only on the y-axis (-Z”). Moreover, you can change the scale to kΩ, then the overall visual will be cleaner. 
9) Line 317: As there are some controversies in the literature, I suggest adding a reference regarding the method the authors applied to calculate the limit of detection.

Author Response

Dear Editor,

Thank you very much for your letter and the referees’ reports. According to your comments and requirements, we have made modification on the original manuscript. Here, we have submitted the revised manuscript online for your approval. A revised manuscript with the correction sections colour marked was attached as the supplemental material and for easy check/editing purpose.

A document of Detailed Responses to Reviewers answering every question from the referees was also summarized and enclosed.

Thank you indeed for your time and efforts.

I’ m looking forward to some good news from you soon.

Best wishes,

Chong Sun

Institute of Agricultural Products Processing

Jiangsu Academy of Agricultural Sciences

Reviewer #2:

The subject presented in the manuscript “A Water-Dispersible Carboxylated-g-C3N4 Nanoparticles-Based Electrochemical Platform for Direct Reporting of Hydroxyl Radical in Meat” is interesting. Therefore, I recommend this manuscript should make minor revision, and some general comments/suggestions/questions are in the following:

1) I recommend not using “g-C3N4” in the title, as it is difficult to read the term out loud. Maybe using “carboxylated carbon nitride nanoparticles” as defined in the Abstract would be a good choice, or something else that authors consider to be a better suit. I have used chemical formulas on my publications before, and I realized later that using a more general term would be easier to promote the paper and therefore increase the citations. Of course, it is just a suggestion based on my previous experience and may not be the rule.

Reply: Thanks for your suggestions. The title was revised according to your suggestion.

“A Water-Dispersible Carboxylated Carbon Nitride Nanoparticles-Based Electrochemical Platform for Direct Reporting of Hydroxyl Radical in Meat”

2) Line 44: Is there a typo on “uarthritis”?

Reply: Thanks for your directions. We have corrected “uarthritis” to “arthritis”.

3) I suggest adding examples of other devices based on electrochemical measurements applied to the detection of the hydroxyl radical in the Introduction section, to complete the state-of-the-art. You may use this opportunity to emphasize what are the advantages of your platform compared to the others in the literature.

Reply: Thanks for your suggestions. We have added related discussion in the section of 1. Introduction.

4) Section 2.5: Is there a reason for conducting CV and EIS in different concentrations (1 mM and 5 mM, respectively)?

Reply: Thanks for your comments. In this paper, CV and EIS were chosen as suitable technique for investigation and monitoring the various stages of the sensors constructed on GCE. The electron transfer between a solution species and the electrode must occur by tunneling either through the barrier or through the defects in the barrier. Different concentrations of [Fe(CN)6]3-/4- used as buffer solution can cause the changes of ionic strength. Higher ionic strength leads to increased charge accumulation in the active channel leading to larger current on the prepared sensor. In this paper, we simply use CV to represent the current change after each assembly step. The current and peak potential in CV plots were definitely different, indicating CV is a convenient way to discuss our sensor only. Thus, 1 mM [Fe(CN)6]3-/4- (1:1) solution containing 0.1 M KCl (pH=7.4) could be applied in CV experiments.

EIS data is generated by applying an AC potential to an electronic device, measuring the AC current response, and recording phase shift and amplitude changes over a range of applied frequencies. Examining the current response over a range of frequencies allows separation of processes which occur on different timescales, making it ideal for separating electronic and ionic processes in modified sensor. In order to amplify the surface properties of different modified sensor, 5 mM [Fe(CN)6]3-/4- (1:1) solution containing 0.1 M KCl (pH=7.4) was chosen to investigate EIS experiments.

5) Line 163: As your paper is intended to be accessed by a global audience, please specify which is the “ambient temperature”.

Reply: Thanks for your suggestions. We have specified the ambient temperature.

“Each test was repeated three times, and all electrochemical measurements were carried out at 25 °C in a digital biochemical incubator.”

6) The figures and table presented in the Supplementary Materials (or Supporting Information, as you named in the word file) would be perfectly fine in the main manuscript. Is there a limited number of figures in the Foods journal? Otherwise, I don’t see a reason why not add them in the main manuscript. The data is not repetitive information and the visual outcome of Figure S5, for instance, would highlight the selective detection of your device towards hydroxyl radical compared to others.

Reply: Thanks for your directions. Some important figures and tables presented in the Supplementary Materials were revised and added into the main manuscript.

7) Line 243: Please define LSV.

Reply: Thanks for your comments. A relevant description has been added into the revised manuscript. Please see Line 250.

8) Figure 3B: On the x-axis, I believe there is no minus sign, it should be positive Z’. Usually, the negative is only on the y-axis (-Z”). Moreover, you can change the scale to kΩ, then the overall visual will be cleaner.

Reply: Thanks for your directions. We have revised Figure 3B according to your suggestion.

Figure 3. (A) CVs of (a) chitosan/GCE, (b) bulk-g-C3N4/chitosan/GCE, (c) carboxylated-g-C3N4/chitosan/GCE and (d) ssDNA/carboxylated-g-C3N4/chitosan/GCE in the 1 mM [Fe(CN)6]3-/4- (1:1) solution containing 0.1 M KCl at a scan rate of 100 mV·s-1, (B) EIS of (a) chitosan/GCE, (b) bulk-g-C3N4/chitosan/GCE, (c) carboxylated-g-C3N4/chitosan/GCE and (d) ssDNA/carboxylated-g-C3N4/chitosan/GCE in the 5 mM [Fe(CN)6]3-/4- (1:1) solution containing 0.1 M KCl (pH=7.4), (C) SWV plots of MB in 0.1 M PBS (pH=7.4) obtained from ssDNA/carboxylated-g-C3N4/chitosan/GCE after incubation with various concentrations of Fenton solution. The concentrations of OH were (a) 4.06 fM, (b) 14.74 fM, (c) 36.14 fM, (d) 48.46 fM, (e) (f) 59.91 fM, (g) 70.34 fM, (h) 78.79 fM, (i) 90.62, (j) 101.89 fM, (k) 113.90 fM and (l) 122.79 fM and (D) The relationship between current difference (△I) and the concentration of OH. △I = I0 - I, I0 and I were the peak currents of MB obtained from electrodes before and after incubated with various concentration of Fenton solution.

9) Line 317: As there are some controversies in the literature, I suggest adding a reference regarding the method the authors applied to calculate the limit of detection.

Reply: Thanks for your comments. A new reference regarding the method to calculate the limit of detection was added in the revised manuscript. Please see 3.5. Electrochemical detection of OH in the revised manuscript.
